# FPTQ: FINE-GRAINED POST-TRAINING QUANTIZATION FOR LARGE LANGUAGE MODELS

## ABSTRACT

In the era of large-scale language models, the substantial parameter size poses significant challenges for deployment. Being a prevalent compression technique, quantization has emerged as the mainstream practice to tackle this issue, which is mainly centered on two recipes W8A8 and W4A16 (i.e. weights and activations in such bit widths). In this study, we propose a novel W4A8 post-training quantization method for the available open-sourced LLMs, which combines the advantages of both two recipes. Therefore, we can leverage the benefit in the I/O utilization of 4-bit weight quantization and the acceleration due to 8-bit matrix computation. Nevertheless, the W4A8 faces notorious performance degradation. As a remedy, we involve layerwise activation quantization strategies which feature a novel logarithmic equalization for most intractable layers, and we combine them with fine-grained weight quantization. Without whistles and bells, we eliminate the necessity for further fine-tuning and obtain the state-of-the-art W4A8 quantized performance on BLOOM, LLaMA, and LLaMA-2 on standard benchmarks. We confirm that the W4A8 quantization is achievable for the deployment of large language models, fostering their wide-spreading real-world applications.

## 1 INTRODUCTION

Large Language Models (LLMs) are distinguished for their exceptional *emergent knowledge capacity* (Wei et al., 2022), enabling them to perform admirably across a wide variety of language tasks. However, their massive scale poses a significant hurdle to deployment due to the substantial storage and the huge amount of computation required. This challenge is particularly pronounced in environments with limited resources such as edge computing devices and personal devices, where the constraints can inhibit the widespread adoption of these cutting-edge language models.

To address this issue, several model compression strategies have been proposed, including pruning (Ma et al., 2023; Frantar & Alistarh, 2023; Sun et al., 2023), distillation (Zhang et al., 2023), quantization (Frantar et al., 2022; Xiao et al., 2023), and low-rank decomposition (Yao et al., 2023). Each of these approaches has its own limitations. For instance, pruning can achieve reasonable compression rates but it may require significant fine-tuning or are closely tied to specific hardware architectures. In contrast, quantization techniques, despite their universal applicability, are often confronted with the problem of significant quantization errors, particularly with the increasing parameter sizes (Dettmers et al., 2022).

Lately, research attention has been shifted towards a more balanced approach to quantization, specifically the usage of lower-bit widths for weights and higher-bit widths for activation, like W4A16 in GPTQ (Frantar et al., 2022). This introduces a novel perspective to tackle the computational and memory-intensive aspects of LLMs, which are typically composed of Transformer Decoder structures (Vaswani et al., 2017). During inference, it can be divided into compute-intensive *context decoding* stage and memory-intensive *self-decoding* stage, each presenting unique challenges and opportunities for further optimization.

However, there is a conspicuous dearth of research that explores the synergistic combination of two quantization recipes W8A8 and W4A16. This paper aims to bridge the gap by proposing an innovative Fine-grained Post-Training Quantization (called FPTQ) method that combines the benefits of both, thereby providing an effective and efficient W4A8 solution for the deployment of a variety of available large language models that are tested across a myriad of natural language tasks.

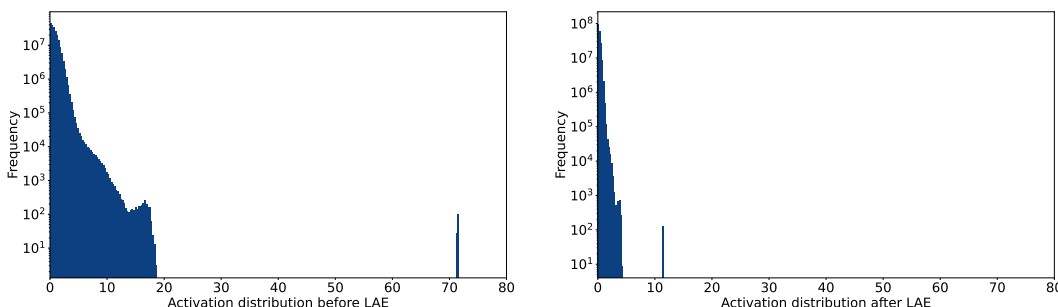

Figure 1: Activation distribution before and after logarithmic equalization on BLOOM-7B1.

We first investigate the quantization difficulty by illustrating the activation distributions in different layers, discovering that their ranges differ dramatically which motivates us for a layerwise strategy. Subsequently, we provide a unique activation equalization technique to handle the intractable outliers (Figure 1), and improve the overall performance with fine-grained weight quantization.

In a nutshell, we make several key contributions to the field of LLM compression and deployment:

1. **High performance and low-cost W4A8 compression**: We are the *first* to achieve high-performance W4A8 (INT4 weights and INT8 activation) PTQ compression for large language models, maintaining the accuracy of the original model. Being a post-training quantization technique, it tremendously simplifies the production flow of LLMs.

2. **Novel quantization scheme**: Based on our comprehensive analysis of the activation distribution of LLMs, we employ a layerwise strategy to cope with different levels of quantization difficulty. Particularly, we devise an offline *logarithmic activation equalization* to render a quantization-friendly distribution for previously intractable layers.

3. **Inference-friendly**: Our approach harmonizes the memory and computation efficiency which enables the storage of weights in a 4-bit format while executing INT8 inference, thereby catalyzing both the memory access and computation.

## 2 RELATED WORK

### 2.1 LARGE LANGUAGE MODELS

The past few years have witnessed the booming of pre-trained language models. BERT (Devlin et al., 2019) is designed to understand the context of words in a sentence and has been used for tasks such as sentiment analysis and question answering. RoBERTa (Liu et al., 2019) is an improved version of BERT with better pre-training techniques and larger training data. T5 (Raffel et al., 2020) is designed to perform a wide range of natural language processing tasks, including language translation and summarization. XLNet (Yang et al., 2019) is designed to handle long sequences of text and has achieved state-of-the-art results on several natural language processing tasks. GPT-3 (Brown et al., 2020) is one of the most advanced LLMs with 175 billion parameters, capable of performing a wide range of natural language processing tasks. Along with the open-sourced ones like GLM (Du et al., 2021), BLOOM (Laurençon et al., 2022), OPT (Zhang et al., 2022) and LLaMa series (Touvron et al., 2023), LLMs have remarkably revolutionized the field of natural language processing and are being used in a wide range of applications.

Nevertheless, LLMs have billions of parameters and are often pre-trained on large amounts of text data, which require significant computational resources to train and deploy. There is a call for faster inference time and lower memory requirements to make LLMs more practical.

## 2.2 QUANTIZATION ON LLMS

Applying quantization to large language models presents unique challenges. Traditional PTQ schemes have achieved great success in Convolutional Neural Networks (CNN) (Nagel et al., 2019; Wu et al., 2020; Nagel et al., 2021; Yao et al., 2021), but direct application to large language models often results in severe accuracy loss, this is typically due to the presence of many outliers in the activation values of large models (Dettmers et al., 2022).

Several approaches have been proposed to address these issues. For example, LLM.int8() (Dettmers et al., 2022) splits the input activation values into two parts: non-outlier dimensions computed with INT8, and outliers computed with FP16. GPTQ (Frantar et al., 2022) and AWQ (Lin et al., 2023) circumvent this difficulty by adopting FP16 activation and INT4 weight-only quantization. However, these methods also have their limitations, such as computational overhead and the inability to truly leverage hardware acceleration.

Other approaches like SmoothQuant (Xiao et al., 2023), RPTQ (Yuan et al., 2023), and ZeroQuant-V2 (Yao et al., 2023) propose different strategies to achieve quantization while mitigating the accuracy loss and computational overhead. However, SmoothQuant is merely a W8A8 solution and it suffers from poor performance on W4A8. The rest tackles the W4A8 challenge but they come with their own set of challenges like weight reordering, asymmetric quantization, and group-wise activation, which can perplex the engineering work and may not well facilitate hardware. In light of these problems, we are driven to achieve W4A8 quantization without relying on QAT or distillation methods, paving the way for the efficient deployment of LLMs.

## 3 METHOD

### 3.1 WHY W4A8?

The generative inference of LLMs can be divided into two stages: *context decoding* that generates an output token given an input prompt (embedded as a sequence of tokens), and *self-decoding* that iteratively predicts the next token in a sequence, see Figure 2 (a). The former is compute-bound due to the first-round computation of lengthy input sequences and the latter is memory-bound as a result of sequential processing, thus two different implementations are required.

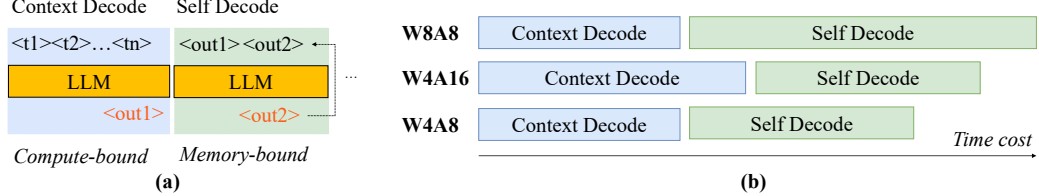

Figure 2: **(a)** Two stages of LLM inference where context decoding is compute-bound and self-decoding is memory-bound. **(b)** W4A8 speeds up both stages and is faster than the other two.

Previous quantization methods like Smoothquant (Xiao et al., 2023) features W8A8, while AWQ (Lin et al., 2023) and GPTQ (Frantar et al., 2022) use W4A16. Both recipes compromise one stage for another, leading to inferior overall performance, whereas only W4A8 can boost both stages, see Figure 2 (b) and Table 1. That is, context decoding enjoys the speed-up using 8-bit matrix multiplication, while self-decoding is also accelerated via reduced memory access using 4-bit weight.

Table 1: Comparison of decoding stage efficiency for different quantization methods. CD: Context Decoding, SD: Self-Decoding

| Method | Efficient CD | Efficient SD |
|---|---|---|
| ZeroQuant | No | No |
| SmoothQuant | Yes | No |
| GPTQ | No | Yes |
| AWQ | No | Yes |
| Ours | **Yes** | **Yes** |

There are a few existing W4A8 studies. ZeroQuant (Yao et al., 2022) utilizes mixed precision for self-attention weights (W8) and is not tested on larger models, ZeroQuantV2 (Yao et al., 2023)

uses fine-grained activation quantization which is not feasible in practice. ZeroQuant-FP (Wu et al., 2023) alleviates the degradation by using higher-precision FP8 computation but it depends on specific hardware (e.g. NVIDIA H100). LLM-QAT (Liu et al., 2023a) adopts QAT to improve W4A8 performance but it requires costly training and is prone to tedious hyper-parameter tuning. Therefore, it is necessary to improve the accuracy of the W4A8 model while not harming its inference speed. The method shall also be made low-cost and generalizable for most up-to-date LLMs.

## 3.2 ANALYSIS OF ACTIVATION DISTRIBUTION ON LLMS

With our goal in mind, we are driven to design a robust PTQ method. To begin with, we study why vanilla W4A8 quantization is difficult for current LLMs. We first draw the activation distribution of LLaMA-7B in Figure 3 to find the distinct behaviors of different layers. For instance, $o_{proj}$ has compact distribution while $down_{proj}$ spans extensively. This phenomenon reoccurs in many other LLMs, see Appendix A.5.

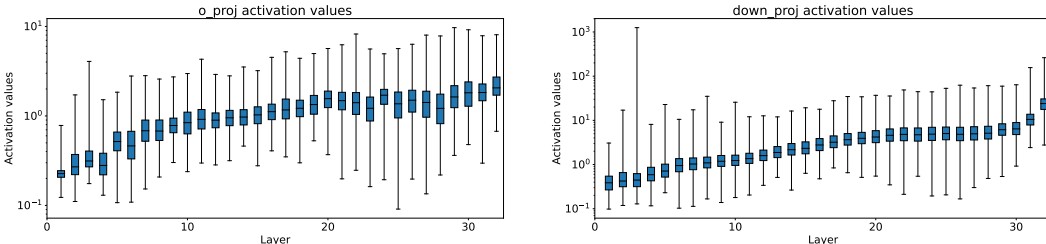

Figure 3: Visualization of activation distribution of $o_{proj}$ and $down_{proj}$ on LLaMA-7B.

As we can see from the above analysis, the maximum fluctuation range of input activation values for certain layers ranges from tens to thousands. Using *per-tensor static quantization* will result in significant quantization errors, but using *per-token dynamic quantization* for all layers will not bring adequate hardware acceleration. Therefore, it naturally calls for a layer-specific policy to determine the granularity of quantization.

## 3.3 FPTQ: FINE-GRAINED POST-TRAINING QUANTIZATION

Motivated by the above analysis, we propose our post-training quantization method which employs a layerwise quantization strategy regarding disparate activation distributions. Our complete procedure is given in Algorithm 1. The key components are discussed in detail.

---

**Algorithm 1** FPTQ: Fine-grained Post-Training Quantization

---
**Input:** A pre-trained LLM
**Output:** A quantized LLM
  1: Calibrate the pre-trained LLM with a predefined dataset
  2: Perform activation distribution analysis
  3: **for each** layer-$l$ in the Transformer structure ($L$ layers in total) **do**
  4:     **if** Activation range $v \leq v_0$ **then**
  5:         Set layer $l$'s activation quantization policy to *static per-tensor*
  6:     **else if** Activation range $v_0 < v < v_1$ **then**
  7:         Perform *logarithmic activation equalization*
  8:         Set layer $l$'s activation quantization policy to *static per-tensor*
  9:     **else**
 10:         Set layer $l$'s activation quantization policy to *dynamic per-token*
 11:     **end if**
 12:     Set each layer's weight quantization policy as fine-grained
 13: **end for**
 14: Update the LLM's weights and activations w.r.t. the chosen quantization policy
 15: Get the high-performance quantized LLM

---

### 3.3.1 LAYER-WISE ACTIVATION QUANTIZATION STRATEGY

The key to resolving the activation quantization difficulty lies in the outlier treatment. Empirically, we can use different activation quantization strategies for different layers, as shown in Table 2. For activation value ranges within tens (denoted as $v_0$), per-tensor static quantization can be safely used. However, to avoid quantization loss for activation ranges over hundreds (denoted as $v_1$), per-token dynamic quantization shall be put in place although slightly sacrificing hardware acceleration benefits. For most layers that range within hundreds, i.e. $(v_0, v_1)$, it demands a particular strategy that simultaneously reduces the quantization error while not harming the inference speed.

Table 2: Activation quantization strategies for different ranges of activation values.

| Activation Value Range | Quantization Strategy | Hardware Efficiency | Typical Operation |
|:---:|:---:|:---:|:---:|
| $v \leq v_0$ | per-tensor, static | High | Dense |
| $v_0 < v < v_1$ | LAE + per-tensor, static | High | QKV, FC1 |
| $v \geq v1$ | per-token, dynamic | Medium | FC2 |

Xiao et al. (2023) discover that when larger outliers dominate the distribution, the effective quantization bits of inliers are substantially narrowed. For per-tensor 8-bit quantization, it becomes $2^8 \cdot m_i/m$ where $m_i$ is the maximum amplitude of channel $i$ and $m$ is the maximum value of the whole tensor. They also observe that outliers stay in fixed channels. Based on these two findings, we are allowed to perform per-channel outlier suppression on activations. SmoothQuant (Xiao et al., 2023) attempts to 'smooth' per-channel distribution by dividing the activation with a scale $s_i = \max(|\mathbf{x}_i|)/\max(|\mathbf{w}_i|)$, where $\mathbf{x}_i$ and $\mathbf{w}_i$ are activation and weight of channel $i$ respectively. AWQ (Lin et al., 2023) introduces grid-searched hyper-parameters $\alpha$ and $\beta$ to lay importance to activation and weight separately, where they find the contribution of weights is marginal and suggest activation-awareness is most important. In this paper, we argue that it is unnecessary to consider weights for computing the activation "smoothing" scale. Besides, it is crucial to retain all the activation values with a *non-linear lossless mapping*, yet it has to satisfy two criteria **(1)** touching gently with the inliers **(2)** harshly suppressing the outliers. In this regard, we verify that the logarithmic function rightly fits this purpose.

**Logarithmic Activation Equalization.** To render a quantization-friendly activation distribution, we propose a new offline *logarithmic activation equalization* (LAE) method that moderates activation distributions in a non-linear fashion. Specifically, we compute the $i$-th channel scale $s_i$ as the maximum activation value $\max(|\mathbf{X}_i|)$ divided by its logarithmic mapping with a shift of 2 (to have a minimum of scale 1), shown in Equation 1. The formula retains the original information while it squashes various distributions comparably. Figure 1 exhibits its outcome distribution.

$$\mathbf{s}_i = \max(|\mathbf{x}_i|)/\log_2(2 + \max(|\mathbf{x}_i|)); \quad \mathbf{x}_i = \mathbf{x}_i/\mathbf{s}_i \tag{1}$$

Once the scale $\mathbf{s}$ is obtained, we can update the corresponding weight and activation as follows,

$$\mathbf{W}' = \text{diag}(\mathbf{s})\mathbf{W}; \quad \mathbf{X}' = \mathbf{X}\text{diag}(\mathbf{s})^{-1} \quad s.t. \quad \mathbf{X}'\mathbf{W}' = \mathbf{X}\mathbf{W} \tag{2}$$

Hence, this update is made *in-place* as it is mathematically equivalent. Notably, $\mathbf{s}$ can be easily fused into the weight of the previous layer. In our case, there are only two types of operations (QKV and FC1) whose activation ranges are in $(v_0, v_1)$. To apply the offline LAE, their activation updates are fused into their preceding operation LayerNorm (Ba et al., 2016).

### 3.3.2 WEIGHT QUANTIZATION

Due to the intricacies of LLMs, it is not tractable to use the vanilla per-channel strategy only, as shown in Figure 4 (a). ZeroQuant (Yao et al., 2022) adopts a fine-grained groupwise weight quantization (Shen et al., 2020) that addresses the quantization difficulty of smaller LLMs like GPT-3 (Brown et al., 2020). As the two strategies are identically costly from the engineering perspective,

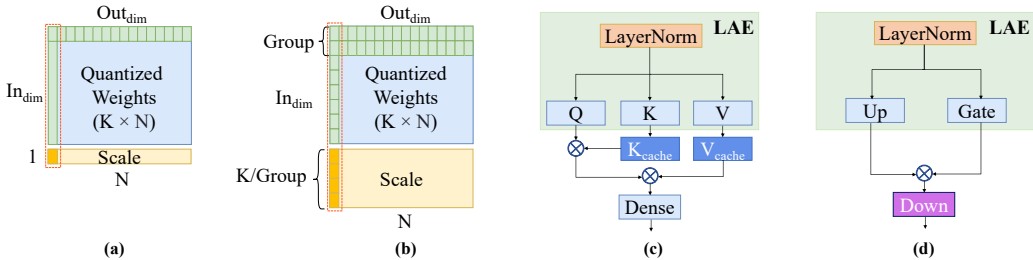

Figure 4: **(a)** Per-channel weight quantization. **(b)** Fine-grained per-channel quantization. **(c, d)** Self-attention and FFN in most LLMs. Light blue: per-tensor static activation quantization. Purple: per-token dynamic activation quantization. All weights are quantized in a fine-grained manner.

we adopt fine-grained weight quantization where the scale is computed groupwise for all layers to obtain better performance, depicted in Figure 4 (b).

As LLaMA series (Touvron et al., 2023) rises to the mainstream focus, we illustrate our specific quantization scheme for its architecture in Figure 4 (c) and (d). Interestingly, we discover that the trend of LLaMA activation distributions holds for all model series, such that our quantization scheme can be directly reused. Logarithmic activation equalization is performed offline (the scale for activation is then fused into LayerNorm) for QKV and Up/Gate. It's also worth noting that the quantized KV cache (a reusable buffer to save the per-tensor static quantization result of KV's activations) is applied to save I/O costs.

# 4 EXPERIMENT

## 4.1 DATASETS

We validated our quantization scheme on several datasets, including LAMBADA (Paperno et al., 2016), MMLU (Hendrycks et al., 2020), and a set of Common Sense QA (Talmor et al., 2019) tasks like WinoGrande (Sakaguchi et al., 2021), PIQA (Tata & Patel, 2003), HellaSwag (Zellers et al., 2019), $ARC_e$. For CommonSense QA tasks, we used the Language Model Evaluation Harness (Gao et al., 2021) tool to evaluate our models. For the calibration set, we randomly sampled 512 samples from the Pile dataset (Gao et al., 2020).

## 4.2 IMPLEMENTATION

**Baselines.** In our experiments, we selected SmoothQuant (Xiao et al., 2023) and GPTQ (Frantar et al., 2022) as our baselines, given their status as the most prevalent W8A8 and W4A16 quantization schemes, respectively. These methods have been widely adopted in various applications and their performance has been extensively validated, establishing them as reliable benchmarks in the field of LLMs quantization. Table 3 exhibits such a comparison with SmoothQuant under various bit widths on the LAM-

| Model | Original FP16 | SmoothQuant W8A8 | W4A8 | FPTQ W4A8 |
|---|---|---|---|---|
| BLOOM-7B1 | 57.9080% | 59.6352% | 58.0245% | 58.2185% |
| LLaMA-7B | 73.7435% | 73.7823% | 64.1762% | 73.8017% |
| LLaMA-13B | 76.1886% | 76.3633% | 69.9010% | 75.7423% |
| LLaMA-65B | 79.1966% | 78.6920% | 69.9787% | 79.1384% |
| LLaMA-2-7B | 73.7046% | 74.1510% | 55.5987% | 72.4820% |
| LLaMA-2-13B | 76.6350% | 75.5288% | 69.7652% | 75.3154% |
| LLaMA-2-70B | 79.5653% | 78.7891% | 76.5185% | 78.7114% |

Table 3: Comparison on the LAMBADA Dataset.

BADA dataset. Notice a vanilla W4A8 version of SmoothQuant suffers from significant degradation. See A.4 for a through component analysis. Simultaneously, to further demonstrate the potential of FPTQ, we compare it with the QAT method, particularly with LLM-QAT (Liu et al., 2023b). It's worth mentioning that QAT introduces a significant computational resource overhead; in contrast, our approach incurs a negligible cost compared to it.

**Implementation.** Based on an empirical analysis (see A.3) of investigated LLMs in our paper, the activation bound $v_0$ can be typically set as 15 and $v_1$ 150.

## 4.3 EXPERIMENTAL RESULTS ON LAMBADA

We initially conducted our experiments on the LAMBADA dataset (Paperno et al., 2016). Despite the fact that LAMBADA may not effectively reflect the comprehensive capabilities of the model, it serves as a valuable tool for rapidly validating model precision and quantifying the impact on model performance. Our method, Fine-grained Post-training Quantization (FPTQ), achieved W4A8 quantized models that demonstrated precision strikingly similar to their floating-point counterparts on both the BLOOM-7B1 (Scao et al., 2022) and all models in the LLaMA series (Touvron et al., 2023). This is a highly encouraging observation, suggesting the efficacy of our approach.

| Model | HyperParam | | MMLU | | | | | Common Sense QA | | | | |
| | Method | BW | Hums. | STEM | Social | Other | Avg | WG | PIQA | HS | $ARC_e$ | Avg |
|---|---|---|---|---|---|---|---|---|---|---|---|---|
| BLOOM-7B1 | FP16 | W16A16 | 26.10 | 26.84 | 24.21 | 26.34 | 25.90 | 63.93 | 72.91 | 57.24 | 57.74 | 62.96 |
| | SmoothQuant | W8A8 | 26.04 | 27.80 | 24.50 | 25.82 | 26.03 | 61.96 | 72.52 | 56.66 | 57.41 | 62.14 |
| | GPTQ | W4A16 | 26.06 | 26.47 | 25.28 | 26.50 | 26.08 | 63.38 | 72.42 | 55.98 | 56.86 | 62.16 |
| | FPTQ | W4A8 | 25.87 | 26.71 | 23.76 | 26.56 | 25.74 | 63.22 | 72.80 | 55.98 | 57.32 | 62.33 |
| LLaMA-7B | FP16 | W16A16 | 33.60 | 31.10 | 38.20 | 38.40 | 35.20 | 69.85 | 79.16 | 76.10 | 72.80 | 74.48 |
| | SmoothQuant | W8A8 | 33.88 | 30.32 | 37.63 | 39.08 | 35.14 | 70.09 | 79.00 | 75.17 | 72.22 | 74.12 |
| | GPTQ | W4A16 | 32.39 | 30.35 | 35.03 | 36.15 | 33.40 | 68.03 | 77.69 | 72.95 | 69.44 | 72.02 |
| | LLM-QAT | W4A8 | 30.0 | 27.4 | 28.4 | 34.2 | 30.0 | 67.7 | 77.5 | 73.5 | 70.2 | 72.22 |
| | FPTQ | W4A8 | 30.20 | 29.95 | 32.76 | 35.87 | 32.02 | 70.01 | 78.40 | 74.46 | 70.79 | 73.42 |
| LLaMA-13B | FP16 | W16A16 | 44.60 | 37.10 | 54.00 | 53.50 | 47.10 | 72.77 | 80.09 | 79.07 | 74.71 | 76.66 |
| | SmoothQuant | W8A8 | 44.14 | 36.51 | 54.05 | 52.65 | 46.64 | 72.06 | 79.71 | 78.34 | 73.91 | 76.00 |
| | GPTQ | W4A16 | 46.01 | 39.00 | 54.01 | 53.36 | 47.96 | 73.16 | 80.25 | 78.60 | 74.37 | 76.59 |
| | LLM-QAT | W4A8 | 38.7 | 32.8 | 47.1 | 47.7 | 41.2 | 70.6 | 79.1 | 77.5 | 73.0 | 75.05 |
| | FPTQ | W4A8 | 40.96 | 34.19 | 49.72 | 49.75 | 43.46 | 72.14 | 79.33 | 77.50 | 72.69 | 75.41 |
| LLaMA-65B | FP16 | W16A16 | 61.80 | 52.00 | 73.30 | 67.60 | 63.50 | 77.35 | 82.32 | 84.15 | 79.76 | 80.90 |
| | SmoothQuant | W8A8 | 61.32 | 50.50 | 71.69 | 66.90 | 62.56 | 74.90 | 81.07 | 82.32 | 77.4 | 78.92 |
| | GPTQ | W4A16 | 60.23 | 52.09 | 72.15 | 66.75 | 62.60 | 77.43 | 82.32 | 83.57 | 79.88 | 80.80 |
| | FPTQ | W4A8 | 59.85 | 49.24 | 71.50 | 65.89 | 61.52 | 75.77 | 81.45 | 83.44 | 78.45 | 79.78 |
| LLaMA-2-7B | FP16 | W16A16 | 43.40 | 37.00 | 51.80 | 52.40 | 46.00 | 69.06 | 79.11 | 75.98 | 74.58 | 74.68 |
| | SmoothQuant | W8A8 | 42.49 | 36.65 | 50.67 | 51.33 | 45.06 | 69.06 | 77.97 | 75.91 | 75.98 | 74.58 |
| | GPTQ | W4A16 | 42.66 | 36.45 | 51.25 | 50.99 | 45.13 | 68.51 | 78.67 | 0.75.96 | 71.68 | 73.45 |
| | FPTQ | W4A8 | 41.15 | 35.79 | 49.37 | 50.77 | 44.02 | 69.38 | 77.97 | 74.89 | 72.85 | 73.77 |
| LLaMA-2-13B | FP16 | W16A16 | 54.40 | 44.30 | 63.40 | 60.80 | 55.70 | 72.22 | 80.52 | 79.38 | 77.44 | 77.39 |
| | SmoothQuant | W8A8 | 52.67 | 43.07 | 63.15 | 60.39 | 54.69 | 72.06 | 79.54 | 79.28 | 77.31 | 77.05 |
| | GPTQ | W4A16 | 51.99 | 43.57 | 63.05 | 60.49 | 54.56 | 72.30 | 79.60 | 78.79 | 77.23 | 76.98 |
| | FPTQ | W4A8 | 51.65 | 42.54 | 62.27 | 59.90 | 53.92 | 70.56 | 79.43 | 78.06 | 75.76 | 75.95 |
| LLaMA-2-70B | FP16 | W16A16 | 65.20 | 57.80 | 80.40 | 74.60 | 69.10 | 77.98 | 82.75 | 83.81 | 80.98 | 81.38 |
| | SmoothQuant | W8A8 | 63.23 | 56.46 | 79.23 | 72.42 | 67.40 | 78.14 | 82.37 | 82.60 | 80.72 | 80.96 |
| | GPTQ | W4A16 | 62.93 | 57.65 | 79.62 | 74.12 | 68.04 | 78.06 | 82.92 | 83.37 | 80.89 | 81.31 |
| | FPTQ | W4A8 | 62.83 | 55.27 | 78.23 | 72.49 | 66.81 | 77.03 | 82.37 | 82.58 | 79.88 | 80.47 |

Table 4: Comparison on MMLU and Common Sense QA. BW: BitWidth

## 4.4 RESULTS ON MMLU AND COMMON SENSE QA

MMLU (Hendrycks et al., 2020) and Common Sense QA (Talmor et al., 2019) are currently renowned datasets that comprehensively reflect the performance of LLMs. We conducted extensive experiments on these datasets, including comparative assessments with three state-of-the-art solutions: SmoothQuant for W8A8, GPTQ for W4A16 and LLM-QAT for W4A8.

On the MMLU dataset, our approach exhibits a performance gap within 1% for most models compared to SmoothQuant. Notable outliers include LLaMA-7B and LLaMA-13B, which show a more pronounced drop. However, it's important to note that the MMLU dataset, with its predominant composition of multiple-choice questions, may exhibit bias in precision estimation when the inherent capabilities of the model are limited.

On Common Sense QA, our approach demonstrates a mere 1% precision gap with the FP16 model across nearly all models, including the previously identified underperforming models LLaMA-7B and LLaMA-13B on MMLU. This observation underscores the robustness of our approach.

## 4.5 COMPARISON WITH LLM-QAT ON COMMON SENSE QA

Given the paucity of other Post-training Quantization (PTQ) works employing W4A8 quantization, we conducted a comparative study with the Quantization-Aware Training (QAT) method, LLM-QAT (Liu et al., 2023b), on the Common Sense QA dataset. Our approach achieved a precision that was notably closer to the FP16 model compared to LLM-QAT. However, due to the limited data publicly available from LLM-QAT, we present here the

| Model | Method | Setting | WG | PIQA | HS | ARC$_e$ | Avg. |
|---|---|---|---|---|---|---|---|
| LLaMA-7B | Original | FP16 | 69.85 | 79.16 | 76.10 | 72.81 | 74.51 |
| | LLM-QAT | W4A8 | 68.80 | 77.40 | 73.00 | 68.40 | 71.90 |
| | FPTQ | W4A8 | **70.09** | **78.62** | **74.45** | **70.37** | **73.38** |
| LLaMA-13B | Original | FP16 | 72.22 | 80.52 | 79.38 | 77.44 | 77.39 |
| | LLM-QAT | W4A8 | 70.60 | 79.10 | 77.50 | 73.00 | 75.05 |
| | FPTQ | W4A8 | **72.85** | **80.09** | **78.20** | **76.09** | **76.81** |

Table 5: Comparison with LLM-QAT on LLaMA-7B.

experimental results for only LLaMA-7B and LLaMA-13B. It can be observed that our approach yields slightly superior results on every subset of the dataset compared to LLM-QAT, highlighting the effectiveness of our methodology.

## 5 ABLATION STUDY

### 5.1 COMPARISON WITH DATA-FREE QUANTIZATION

We acknowledge that the calibration dataset may be one of the factors affecting the performance of the quantized model. Therefore, to maintain fairness, we utilized the Pile dataset (Gao et al., 2020) as a calibration dataset in our previous experiments. However, to demonstrate the robustness of our method, we applied randomly generated tokens for model calibration. We conducted ablation studies on BLOOM-7B1, LLaMA-7B and LLaMA-2-7B under W8A8 and W4A8 bit-width settings in Table 6. It's exhilarating to note that, it was found that using a random dataset often resulted in superior results in most cases. This attests that our method is also applicable in data-free situations.

| Model | HyperParam Calibration | BW | MMLU Hums. | STEM | Social | Other | Avg | Common Sense QA WG | PIQA | HS | ARC$_e$ | Avg |
|---|---|---|---|---|---|---|---|---|---|---|---|---|
| LLaMA-7B | Pile | W8A8 | 33.88 | 30.32 | 37.63 | 39.08 | 35.14 | 70.09 | 79.00 | 75.17 | 72.22 | 74.12 |
| | Random | W8A8 | 32.33 | 29.85 | 36.46 | 38.25 | 34.07 | 70.01 | 78.62 | 75.48 | 72.69 | 74.20 |
| | Pile | W4A8 | 30.20 | 29.95 | 32.76 | 35.87 | 32.02 | 70.01 | 78.40 | 74.46 | 70.79 | 73.42 |
| | Random | W4A8 | 31.20 | 31.05 | 36.37 | 37.01 | 33.64 | 68.67 | 78.62 | 74.62 | 71.21 | 73.28 |
| LLaMA-2-7B | Pile | W8A8 | 42.49 | 36.65 | 50.67 | 51.33 | 45.06 | 69.06 | 77.97 | 75.91 | 75.98 | 74.58 |
| | Random | W8A8 | 42.55 | 36.28 | 51.41 | 51.63 | 45.24 | 67.80 | 79.22 | 75.98 | 74.28 | 74.32 |
| | Pile | W4A8 | 41.15 | 35.79 | 49.37 | 50.77 | 44.02 | 69.38 | 77.97 | 74.89 | 72.85 | 73.77 |
| | Random | W4A8 | 41.32 | 35.42 | 47.97 | 50.19 | 43.56 | 67.88 | 78.07 | 75.46 | 73.11 | 73.63 |
| BLOOM-7B1 | Pile | W8A8 | 26.04 | 27.80 | 24.50 | 25.82 | 26.03 | 63.93 | 72.91 | 57.24 | 57.74 | 62.96 |
| | Random | W8A8 | 25.80 | 27.60 | 25.06 | 26.96 | 26.29 | 63.77 | 72.80 | 56.65 | 57.45 | 62.67 |
| | Pile | W4A8 | 25.87 | 26.71 | 23.76 | 26.56 | 25.74 | 61.96 | 72.52 | 56.66 | 57.41 | 62.14 |
| | Random | W4A8 | 26.29 | 27.04 | 23.37 | 27.08 | 25.99 | 61.88 | 72.42 | 56.17 | 56.94 | 61.85 |

Table 6: Ablation study on calibration datasets on MMLU and Common Sense QA.

### 5.2 WEIGHT QUANTIZATION WITH GPTQ

We observe that the GPTQ method, which compensates weights based on the Hessian matrix, is orthogonal to our existing approach. Therefore, we attempted to fine-tune the weights using the GPTQ method after conducting logarithmic activation equalization (LAE) on the model, to investigate the potential for increased precision. However, our experiments in Table 7 demonstrated that the addition of GPTQ operations generally resulted in a negative impact on precision in most cases. We encourage future researchers to conduct more intriguing explorations in this area.

| Model | HyperParam | | MMLU | | | | | Common Sense QA | | | | |
| | Method | BW | Hums. | STEM | Social | Other | Avg | WG | PIQA | HS | ARC$_e$ | Avg |
|---|---|---|---|---|---|---|---|---|---|---|---|---|
| | FP16 | W16A16 | 33.60 | 31.10 | 38.20 | 38.40 | 35.20 | 69.85 | 79.16 | 76.21 | 72.81 | 74.51 |
| LLaMA-7B | FPTQ | W4A8 | 30.20 | 29.95 | 32.76 | 35.87 | 32.02 | 70.01 | 78.40 | 74.46 | 70.79 | 73.42 |
| | FPTQ$_{GPTQ}$ | W4A8 | 28.40 | 28.33 | 30.84 | 33.22 | 30.03 | 68.82 | 78.13 | 72.88 | 66.96 | 71.70 |
| | FP16 | W16A16 | 43.40 | 37.00 | 51.80 | 52.40 | 46.00 | 69.06 | 79.11 | 75.98 | 74.58 | 74.68 |
| LLaMA-2-7B | FPTQ | W4A8 | 41.15 | 35.79 | 49.37 | 50.77 | 44.02 | 69.38 | 77.97 | 74.89 | 72.85 | 73.77 |
| | FPTQ$_{GPTQ}$ | W4A8 | 40.57 | 35.42 | 48.55 | 48.86 | 43.13 | 67.56 | 78.35 | 74.90 | 72.94 | 73.44 |
| | FP16 | W16A16 | 26.10 | 26.84 | 24.21 | 26.34 | 25.90 | 63.93 | 72.91 | 57.24 | 57.74 | 62.96 |
| BLOOM-7B1 | FPTQ | W4A8 | 25.87 | 26.71 | 23.76 | 26.56 | 25.74 | 61.96 | 72.52 | 56.66 | 57.41 | 62.14 |
| | FPTQ$_{GPTQ}$ | W4A8 | 26.21 | 28.20 | 25.28 | 26.37 | 26.47 | 62.90 | 72.31 | 55.39 | 57.28 | 61.97 |

Table 7: Ablation on MMLU and Common Sense QA. FPTQ$_{GPTQ}$: weights updated by GPTQ first.

## 6 DISCUSSION AND FUTURE DIRECTIONS

**Analysis on computation efficiency.** Modern GPUs, such as the NVIDIA A100, support parallel block-wise matrix computation and pipeline processing. Fine-grained weight quantization enjoys such block-wise computation and introduces little overhead. Currently, the W4A16 acceleration is based on the GPU `FP16INT4 GEMM` kernel (Kim et al., 2022), which implements mixed-type matrix computation. The INT4 weights are first converted to FP16, and matrix computation is then performed with FP16. The underlying computation still uses the GPU's floating-point computation unit, so in the case of long inputs and large batches, the `FP16INT4` kernel even has a negative effect compared to direct FP16 computation because of the additional conversion. The W8A8 computation acceleration is based on the GPU `INT8 GEMM` kernel, which uses INT8 Tensor Cores for underlying computation. There is a noticeable acceleration in the context decoding stage, but in the self-decoding stage, the bottleneck mainly lies in memory access.

To simultaneously address the acceleration issues in both the context decoding and self-decoding stages, we can design an `INT8INT4` kernel, which profits INT8 Tensor Cores for acceleration in the context decoding stage, while keeping the weights loaded as INT4 to reduce memory access time in the self-decoding stage.

**Data-free quantization.** We discover that it is promising to randomly draw samples from the token vocabulary as in Table 6. We believe that there is still room for improvement in this regard.

**Scale computation requires activation only.** For activation quantization, our method completely removes weights for the computation of $s_i$ which echoes the findings in Lin et al. (2023). To make our strategy more generalizable, we introduce a hyper-parameter $\alpha$ to control the level of suppression, see A.2. It is however possible to devise other non-linear mapping functions that are hyper-parameter free and in the meanwhile lead to better performance.

## 7 CONCLUSION

In conclusion, our work presents a significant stride in the domain of Large Language Model (LLM) compression. Upon an overview of the existing quantization schemes, we introduce a novel post-training quantization approach that can make the inference of LLMs more efficient, without compromising their performance. We successfully achieved high performance and efficiency for W4A8, which has the optimal utilization of computational resources which enhances the speed of both content-decoding and self-decoding stages. Furthermore, the removal of the need for fine-tuning during the training process simplifies the deployment pipeline significantly. This attests that our method provides an effective deployable solution for LLMs without sacrificing their accuracy. While our progress is encouraging, we acknowledge the potential for further exploration and refinement in this area. We anticipate that our work will inspire future research endeavors aimed at making LLMs even more efficient and practical.

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

## A    APPENDIX

### A.1    PRELIMINARY KNOWLEDGE ON QUANTIZATION

Quantization is a process of mapping continuous values to discrete ones by scaling. The scaling factor is also called quantization step size. In practice, a higher-precision floating point is used for training and the quantized version is used for inference. Consider $b$-bit integer quantization, for a real tensor $\mathbf{x}$ ranges in $(min, max)$, it can be converted to an integer tensor $\mathbf{x}'$ within $(-2^{b-1}, 2^{b-1} - 1)$ by symmetric uniform quantization as,

$$scale = max(|\mathbf{x}|)/(2^{b-1} - 1) \tag{3}$$

$$\mathbf{x}' = \lfloor (\mathbf{x}/scale) \rceil \tag{4}$$

**Weight quantization and activation quantization.** Typically, the weight is quantized as integer values. Activation quantization refers to the quantization of intermediate activation feature maps.

**Static quantization vs. dynamic quantization.** For static quantization, offline activation statistics are collected to compute the scale and it is kept static during inference. For dynamic quantization, such statistics are computed at runtime.

**Per-tensor vs. per-token.** In the per-tensor scheme, the tensor matrix is considered as a whole to compute the quantization scale. In the per-token scheme, each input token corresponds to a scale computed upon all activation channels of the specific token. In essence, the per-token scheme is more fine-grained.

**Per-channel vs. group-wise.** In the per-channel scheme, the quantization scale is computed channel-wise. In the group-wise scheme, each channel is divided into several groups and so are its scales.

### A.2    GENERALIZED FORM OF LAE

We give a generalized form of logarithmic activation equalization function. For the majority of LLMs, we use $\alpha = 1$.

$$scale = (\log_2(2 + scale))^{\alpha} \tag{5}$$

### A.3    ACTIVATION DISTRIBUTION ANALYSIS ON TYPICAL LAYERS

For the investigated LLMs in our paper, we calculate the activation ranges per layer type in Table 8. Notice that the ranges of QKV and FC1 create difficulties for activation quantization.

| Model | QKV | Dense | FC1 | FC2 |
|---|---|---|---|---|
| OPT-66B | 3.2871/94.87 | 0.0581/16.45 | 3.3223/94.81 | 0.1580/34.00 |
| BLOOM-176B | 1.7451/97.31 | 0.0972/17.07 | 3.9922/119.8 | 0.1636/71.00 |
| GLM-130B | 2.7188/115.1 | 0.2861/19.39 | 3.3125/124.5 | 0.5513/5476. |
| LLaMA-65B | 0.4927/22.51 | 0.0518/17.40 | 0.2856/27.07 | 0.0485/887.5 |

Table 8: Activation range (min/max) of typical layers in LLMs.

After the application of our proposed logarithmic activation equalization on QKV and FC1, we have a smoothed activation range that is much easier to quantize, see Table 9.

### A.4    COMPONENT ANALYSIS ON SMOOTHQUANT W4A8

| Model | QKV | Dense | FC1 | FC2 |
|---|---|---|---|---|
| OPT-66B | 0.2637/4.03 | 0.0572/16.45 | 0.3489/3.28 | 0.1580/34.00 |
| BLOOM-176B | 0.2446/4.66 | 0.0972/17.07 | 0.4021/6.46 | 0.1636/71.06 |
| GLM-130B | 0.4658/1.89 | 0.2861/19.37 | 0.4006/2.01 | 0.5513/5476. |
| LLaMA-65B | 0.2371/3.10 | 0.0518/17.39 | 0.1586/3.69 | 0.0485/887.5 |

Table 9: Smoothed activation range (min/max) of typical layers in LLMs.

To illustrate the effect of each component of our proposed approach, we apply it on the vanilla W4A8 adaption of SmoothQuant. The result is shown in Table 10. We observe that per-tensor-only quantization renders a notorious performance collapse. Applying per-token for activation significantly reduces the error. When combined with fine-grained weight quantization and per-token activation quantization only for FC2 , the gap further decreases. Finally, our proposed FPTQ, which is an amalgamation of all these components along with renovated logarithmic activation equalization, achieves the state-of-the-art W4A8 performance on LAMBADA dataset.

| Method | LLaMA-1-7B | LLaMA-1-13B | LLaMA-1-65B | LLaMA-2-7B | LLaMA-2-13B | LLaMA-2-70B |
|---|---|---|---|---|---|---|
| FP16 | 73.7435 | 76.1886 | 79.1966 | 73.7046 | 76.6350 | 79.5653 |
| SQ + pts | 30.8946 | 47.8944 | 52.3190 | 17.0192 | 28.5171 | 64.9136 |
| SQ + ptk | 64.1762 | 69.9010 | 69.9787 | 55.5987 | 69.7652 | 76.5185 |
| SQ + fg + FC2 ptk | 72.9090 | 76.0916 | 78.5368 | 71.5311 | 74.1898 | 78.5950 |
| FPTQ | 73.8017 | 75.7423 | 79.1384 | 72.4820 | 75.3154 | 78.7114 |

Table 10: Component analysis vs. SmoothQuant (SQ) W4A8 on the LAMBADA dataset. fg: fine-grained, pts: per tensor, ptk: per-token

## A.5 MORE ACTIVATION DISTRIBUTION OF LLMS

We visualize the activation distributions of the LLaMA series in Figure 5, 6, 7, 8, 9, and 10. It is rather exciting to find that LLaMA models at different scales share similar distributions in the same operations, which leads to a universal quantization scheme.

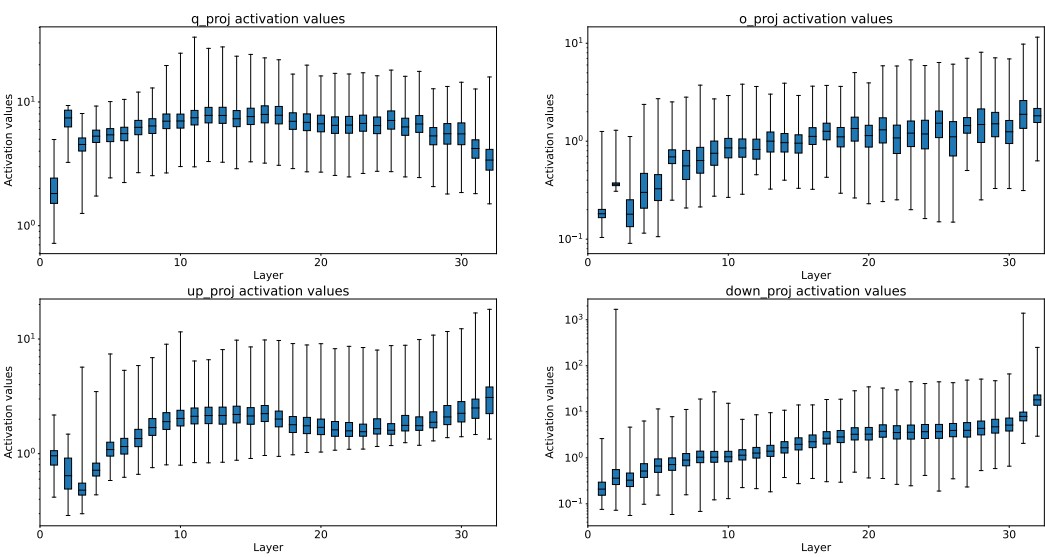

Figure 5: Visualization of activation distribution of $o_{proj}$ and $down_{proj}$ on LLaMA-2-7B.

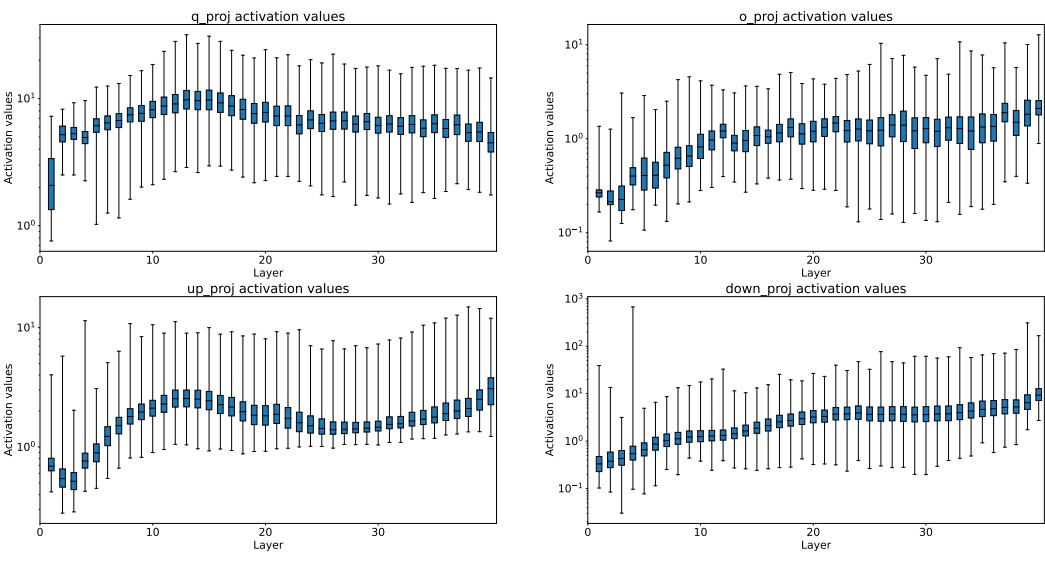

Figure 6: Visualization of activation distribution of $o_{proj}$ and $down_{proj}$ on LLaMA-2-13B.

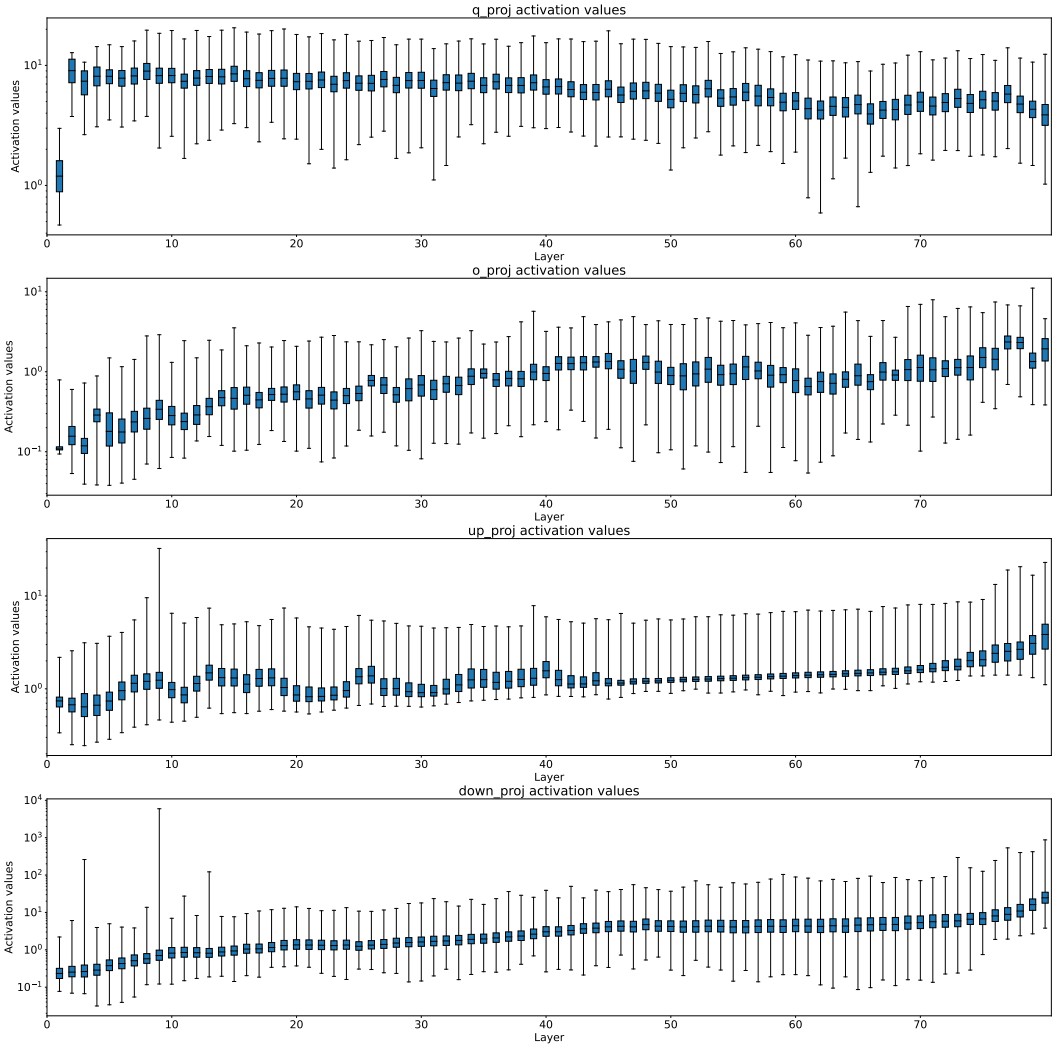

Figure 7: Visualization of activation distribution of $o_{proj}$ and $down_{proj}$ on LLaMA-2-70B.

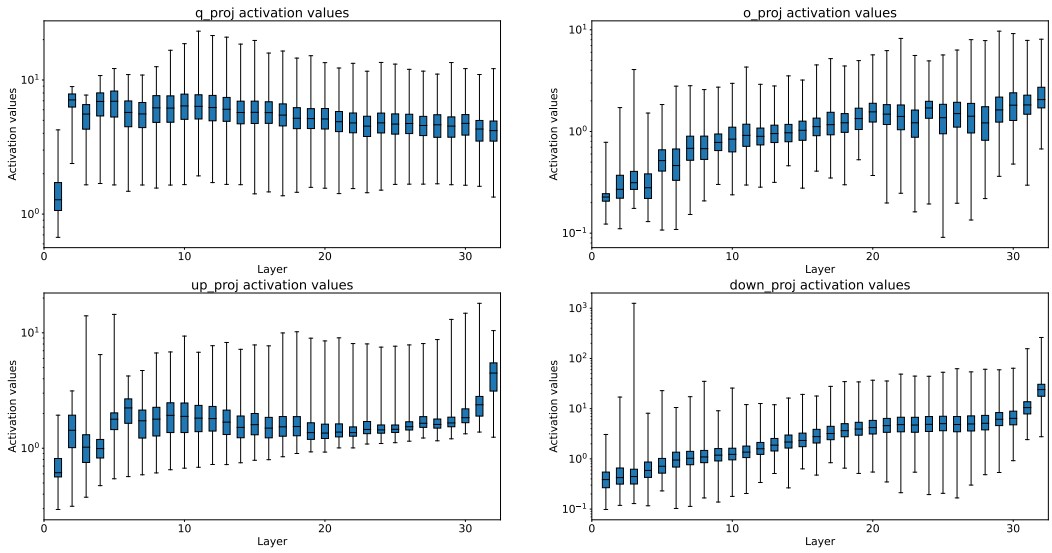

Figure 8: Visualization of activation distribution of $o_{proj}$ and $down_{proj}$ on LLaMA-7B.

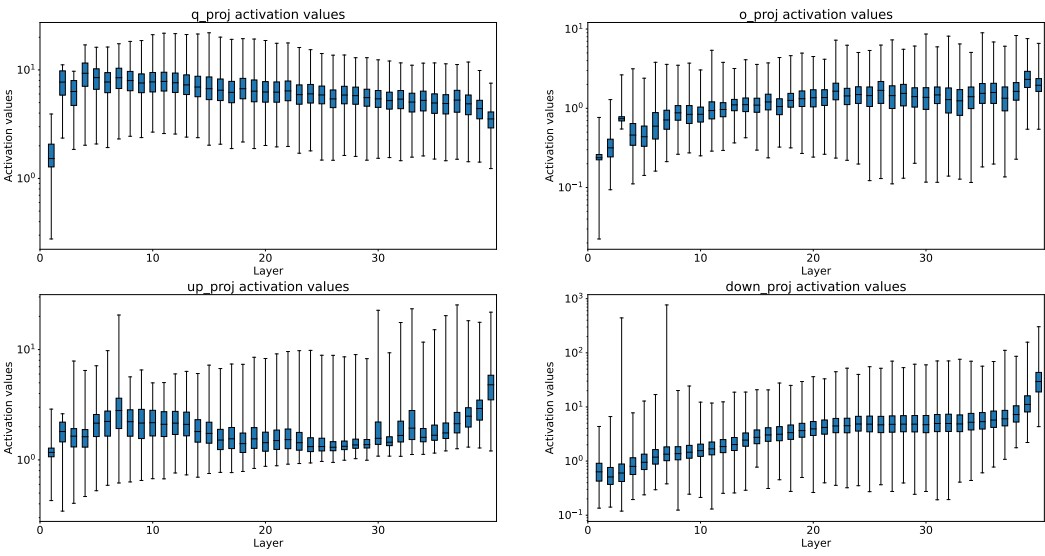

Figure 9: Visualization of activation distribution of $o_{proj}$ and $down_{proj}$ on LLaMA-13B.

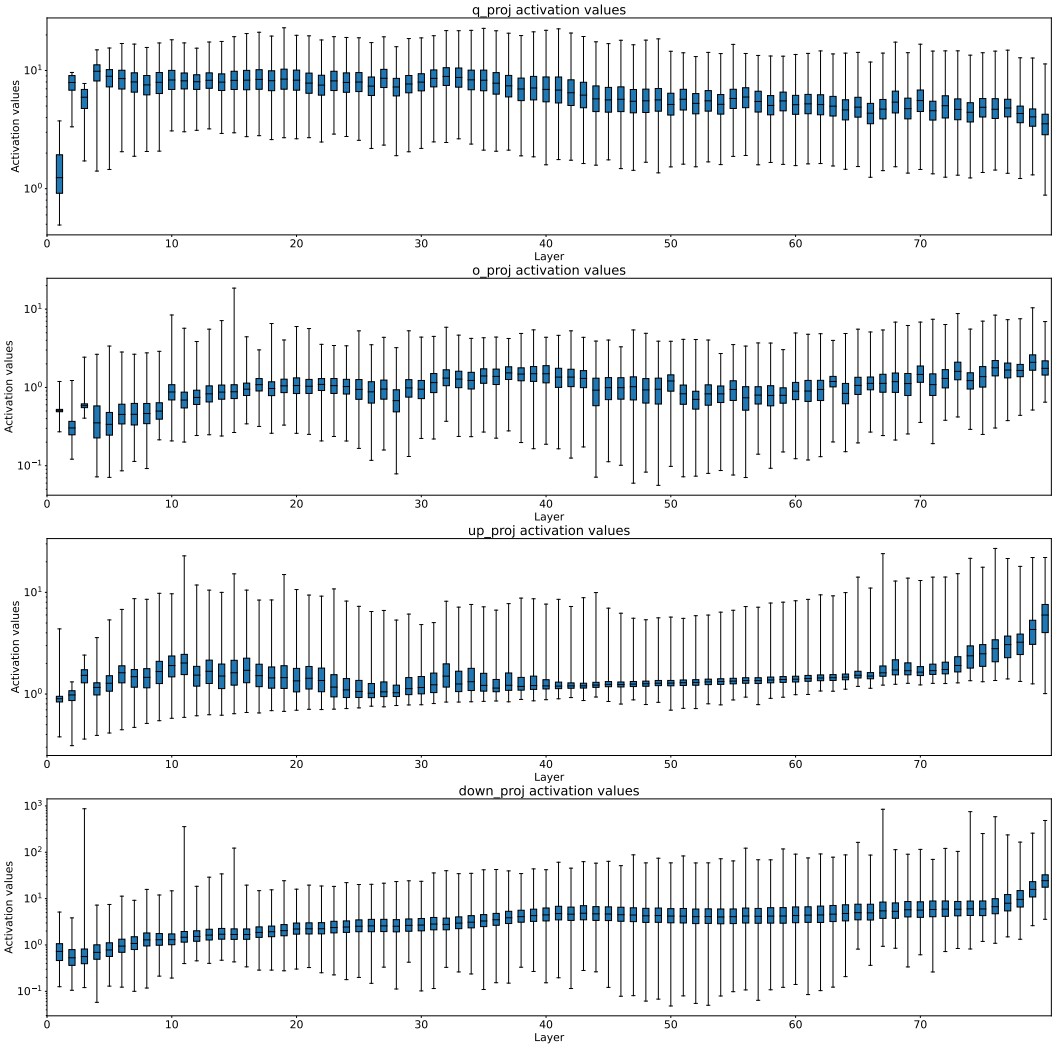

Figure 10: Visualization of activation distribution of $o_{proj}$ and $down_{proj}$ on LLaMA-65B.

