# OpenReview forum: "FPTQ: FINE-GRAINED POST-TRAINING QUANTIZATION FOR LARGE LANGUAGE MODELS"
_ICLR.cc/2024/Conference — Submitted to ICLR 2024_

### Official Review · Reviewer_uu7J · 2023-10-31

**Soundness:** 2 fair
**Presentation:** 2 fair
**Contribution:** 1 poor
**Rating:** 3
**Confidence:** 4

**Summary:**

This study introduces a novel post-training quantization method, termed FPTQ, which effectively combines logarithmic equalization and layer-wise activation quantization for challenging layers, along with fine-grained weight quantization (W4A8). FPTQ establishes the practicality of W4A8 quantization for deploying large language models.

The paper underscores the significance of W4 → W8 dequantization for acceleration through INT8-GEMM in the summarization phase. Furthermore, it highlights the potential adoption of W4A8 in the weight-only quant kernel during the generation phase, emphasizing the preference for INT8-GEMM over FP16-GEMM.

**Strengths:**

* This paper introduces the innovative W4A8 quantization method, marking the first utilization of the PTQ approach.
* The proposed method is validated through extensive experimentation involving diverse model sizes and datasets.
* The robustness of the proposed approach is demonstrated through ablation studies conducted on the calibration set.
* Notably, it showcases superior accuracy with reduced data and time requirements when compared to alternative QAT methods.

**Weaknesses:**

* In assessing the capabilities of the LLM model, the MMLU score holds significant importance. Nevertheless,  the degradation of the FPTQ MMLU score is significant, even when accounting for variations in bit precision relative to existing methods.
* The proposed approach, considering the inference characteristics observed on GPU during the context-decoder and self-decoder stages, suggests the effectiveness of W4A8. However, it is important to acknowledge that this observation might be applicable to a broad spectrum of quantization papers employing W4A8. Therefore, when assessing the significance of Fine-Grained, as emphasized in the paper, it may not be deemed a robust contribution from the perspective of the paper's overall contributions.
* The utilization of LAE necessitates the definition of values for $v_0$ and $v_1$, a process that seems somewhat heuristic and potentially time-consuming to identify optimal settings. Moreover, the sensitivity of $v_0$ and $v_1$ to alterations remains unclear.

**Questions:**

* The Context-Decoding stage appears to be compute-bound, which may limit the I/O benefits of W4A8, and the additional overhead during the conversion from W4 to W8 could potentially lead to longer Context-Decoding times compared to W8A8.
* How is the determination of the clipping value managed when employing activation static quantization?
* Has KV cache quantization been implemented in the process?
* It would be valuable to conduct a comparative analysis of LLM-QAT using the MMLU score as a reference.
* An ablation study would provide insights into the sensitivity of the $v_0$ and $v_1$ values, which is currently unclear.
* The significance of the bold numbers in Table 4 requires clarification.
* Typo alert: In Table 5, HS, LLaMA-7B original accuracy seems to be exceptionally low.

---

> ### Author Response · Authors · 2023-11-19
> **FPTQ prevails over LLM-QAT W4A8 on MMLU scores**
>
> $\textbf{Q1:}$  Degradation of the MMLU score and comparison with LLM-QAT.
>
> $\textbf{A1:}$ Indeed, there is an observable degradation of the MMLU score. Considering we apply a more aggressive quantization (W4A8), a certain degree of performance degradation is reasonable and expected. A more severe degradation is seen in LLM-QAT. We quote their reported MMLU scores (in their Appendix) here for a fair comparison. Note under the same bit width setting W4A8, our PTQ method FPTQ outperforms their costly QAT method by a clear margin.
>
> | Method | Model  | Humanities | STEM | Social Sciences |  Other | Average |
> |:------: | :------:|:------:|:------:|:------:|:------:|:------:|
> | LLM-QAT | LLaMA-1-7B | 30.0 |27.4 |28.4 |34.2 |30.0|
> | FPTQ | LLaMA-1-7B | **30.20** |  **29.95**  | **32.76** |  **35.87** |  **32.02**  |
> | LLM-QAT | LLaMA-1-13B |38.7 |32.8| 47.1 |47.7 |41.2 |
> | FPTQ | LLaMA-1-13B |**40.96**  | **34.19**  | **49.72**  | **49.75** |  **43.46** |
>
> *Table v. Comparison with LLM-QAT W4A8 on MMLU*
>
> It's important to highlight that our MMLU scores are very close to those achieved with W8A8 or W4A16 quantization. We believe this is a testament to the effectiveness of our approach. Furthermore, it's worth noting that existing methods can experience significant performance drops even with W8A8 or W4A16 quantization on some sensitive models. The trade-off here is the significantly improved computational efficiency, which is a key goal of our work.
>
> $\textbf{Q2:}$ Significance of fine-grained quantization.
>
> $\textbf{A2:}$ We concur with your viewpoint that the observed inference characteristics could be applicable to a broad range of quantization papers utilizing W4A8. Nonetheless, our emphasis on fine-grained quantization stems from its unique implementation in our work and its contribution to the overall effectiveness of our approach. We would like to underscore that while fine-grained quantization can help reduce accuracy loss (see also $\textbf{A1}$ to Reviewer yeXC), it can potentially lead to a significant decrease in inference speed as well (see also Analysis on computation efficiency in Section 6).
>
> $\textbf{Q3:}$ The selection v0 and v1 and their sensitivy.
>
> $\textbf{A3:}$ They are empirical values, see also $\textbf{A3}$ to Reviewer WDjT. Note that v0 and v1 serve as a ruler to select which layers require advanced quantization techniques. According to our statistical analysis over a large body of LLMs, the two numbers can be mostly kept the same.
>
> $\textbf{Q4:}$ Cost of context-decoding stage.
>
> $\textbf{A4:}$ Type conversion does add extra cost but it also depends on the implementation. We have revised Figure 2 (b) for a more accurate depiction.
>
> $\textbf{Q5:}$ How to determine clipping value when employing activation static quantization?
>
> $\textbf{A5:}$ In our experiments, we don't apply clipping for activation. As for LAE in Eq.1, we use min/max for smoothing.
>
>
> $\textbf{Q6:}$ Has KV cache quantization be implemented?
>
> $\textbf{A6:}$ Yes, we apply the KV cache as a reusable buffer to save the per-tensor static quantization result of KV’s activations.
>
> $\textbf{Minors}$
>
> Typos are fixed, see also our common response.

---

> > ### Author Response · Authors · 2023-11-21
> > **Any Further Questions?**
> >
> > Dear Reviewer uu7J,
> >
> > We are desperate to get to know your response. We are enthusiastic to address any further questions raised within the precious rebuttal period. It will be crucial for us to refine this paper to well serve the LLM compression community.
> >
> > Best regards,
> >
> > The Authors

---

> ### Comment · Reviewer_uu7J · 2023-11-23
>
> Thank you for the detailed answers and results.
> I have read the authors' rebuttal as well as other reviews. I would like to keep my rating.

---

### Official Review · Reviewer_yeXC · 2023-10-31

**Soundness:** 1 poor
**Presentation:** 1 poor
**Contribution:** 2 fair
**Rating:** 3
**Confidence:** 5

**Summary:**

This paper proposes a SmoothQuant-like approach to address the well-known "channel-specific outlier problem" for LLMs, i.e. some specific channels in activations tend to have much larger range compared to the others, which will cause difficulty in choosing quantization scaling factor and degradation in accuracy. The proposed method chooses the quantization scaling factor based on a log function of the max activations solely, i.e. as opposed to using both activations and weights in SmoothQuant. This method will be applied only when the range of activations of a given layer is between 15 and 150, above which the quantization scheme will fallback to per-token dynamic approach. The author then demonstrated the proposed method using W4A8 (INT4 and INT8) quantization settings and benchmarked with W8A8 (using SmoothQuant) and W4A16 (GPTQ FP16 matrix multiplication), showing comparable accuracy. The author argues that W4A8 would be a better combination than the other two cases, as the compute-intense stage in LLM could benefit from INT computations and the memory-bound stage in LLM could still enjoy the reduced communication due to INT4 weights.

**Strengths:**

1. The proposed method is simple and straightforward to implement.
2. sufficient experiments on different sizes of LLaMA family.

**Weaknesses:**

1. Novelty. The proposed method is very similar to SmoothQuant, main difference is the formula to calculate the scaling factor. Not only it seems like a variation from SmoothQuant, but it didn't show enough proofs that the proposed formula is superior than the original one. For example, the author doesn't show any W4A8 SmoothQuant results and contrast with the "failed SmoothQuant."

2. Unsupported claims. For example, it seems like the author didn't implement the W4A8 INT kernels for a real inference speed test, which makes Fig 2b a speculation rather than real data (which would be ok). However, this plot doesn't seem to be reasonable. If W4A8 is using the same 8bit INT GEMM engine as W8A8, it would require some additional conversion for W4 to W8 (similar for W4A16 case.) That means under compute-bound assumption, the best the "blue box" could do would be very close to W8A8, assuming the conversion overhead can be nicely overlapped with computation. Similar for the memory-bound portion compared to W4A16. This plot seems to exaggerate the benefit of W4A8 without solid support.

**Questions:**

1. A few details in the experiments seems to be missing. For example, in Table 4, how bad is SmoothQuant at W4A8? If SmoothQuant doesn't perform well, was it because the scaling factors or was it the layers need to be treated in the same way as in FPTQ, such that some FC2s are using dynamic per-token settings? And how was v0 and v1 chosen? What's the impact of those parameters? As a quantization method paper, these details would help the readers to build better understanding about the proposed method.

2. The two main claims of this work are a) W4A8 with outlier treatment can work and b) W4A8 can bring new benefit in terms of inference speed. Considering there are a few works trying to tackle the same issue at even lower precisions, e.g. OmniQuant and QLLM shows W4A4 results, together with all the W8A8 works and W4A16 works..., if the second claim can be demonstrated with real hardware results, it would make this paper much stronger. If the author has any progress and results on that front and willing to share, please consider to add them to the manuscript.

---

> ### Author Response · Authors · 2023-11-19
> **FPTQ substantially improves the vanilla SmoothQuant at W4A8**
>
> $\textbf{Q1:}$ Comparison with SmoothQuant at W4A8 and further component analysis.
>
> $\textbf{A1:}$ We give the comparison and the step-by-step ablation in Table iv below. A vanilla SmoothQuant W4A8 is much inferior. In contrast, our proposed FPTQ substantially resolves the degradation in the W4A8 setting. We emphasize that FPTQ takes full advantage of fine-grained weight quantization, a layer-wise activation quantization strategy, and logarithmic activation equalization, altoghther to have on-par or better performance compared with SmoothQuant W8A8 in Table 3.
>
> | Method | LLaMA-1-7B | LLaMA-1-13B | LLaMA-1-65B | LLaMA-2-7B |  LLaMA-2-13B |  LLaMA-2-70B |
> | ---- | ------| -----| -----|-----|------|------|
> | SmoothQuant W4A8 + All Per-tensor |  30.8946% | 47.8944% | 52.3190% | 17.0192%| 28.5271% | 64.9136% |
> | SmoothQuant W4A8 + All Per-token |  64.1762% | 69.9010% | 69.9787% | 55.5987% | 69.7652% | 76.5185% |
> | SmoothQuant W4A8 + Fine-grained +  FC2 Per-token  |  72.9090% | 76.0916% | 78.5368% | 71.5311% | 74.1898% | 78.5950% |
> | FPTQ W4A8 | 73.8017% | 75.7423% | 79.1384% | 72.4820% | 75.3154% | 78.7114% |
>
> *Table iv: SmoothQuant vs. FPTQ strategies on the LAMBADA dataset.*
>
> These results are now updated in the revised manuscript.
>
> $\textbf{Q2:}$ Concerned on the time cost plot of W4A8 vs. W8A8 and W4A16.
>
> $\textbf{A2:}$ Indeed, during the context decoding stage when the computation bound is met, the W4A8 and W8A8 would have close computation cost theoretically. However, it also depends on practical input lengths where W4A8 enjoys I/O benefit, especially at shorter input lengths. Similarly, W4A8 should share a similar time cost with W4A16 in the self-decoding stage. This figure is redrawn to be more accurate.
>
> $\textbf{Q3:}$ How was v0 and v1 chosen?
>
> $\textbf{A3:}$ See $\textbf{A3}$ to Reviewer WDjT.
>
> $\textbf{Q4:}$ Real hardware results.
>
> $\textbf{A4:}$ We would be very glad to share when it gets ready. However, we focus on a theoretical upper bound here, see also  $\textbf{A1}$ to Reviewer WDjT.

---

> > ### Author Response · Authors · 2023-11-21
> > **Have we addressed your concerns?**
> >
> > Dear Reviewer yeXC,
> >
> > As the rebuttal period almost comes to an end, we are eager to know whether our response well addresses your concerns. In case there are any further issues, we are more than willing to answer them with our best effort.
> >
> > Best regards,
> >
> > The Authors

---

> > > ### Comment · Reviewer_yeXC · 2023-11-21
> > >
> > > My review rating for this paper would stay the same. Thanks for all the answers and the interesting experimental results on SmoothQuant W4A8.

---

### Official Review · Reviewer_WDjT · 2023-10-31

**Soundness:** 3 good
**Presentation:** 3 good
**Contribution:** 2 fair
**Rating:** 6
**Confidence:** 4

**Summary:**

The authors propose a new quantization method for LLMs which can utilize both 4-bit weights for memory bandwidth saving and 8-bit activations for faster computation (W4A8). Especially, it's known that naive W4A8 method degrade model performance significantly. The authors use group-wise quantization for the model weights, and Logarithmic Activation Equalization (LAE) for the activations. In particular, it is challenging to quantize activations, and the authors use different strategies on different kinds of layers based on the statistics observed. For those strategy selections, the proposed method uses some calibration datasets. As a result, the author could achieve comparable quantization performance with W4A8. The authors show this on various LLMs including BLOOM, LLaMA and LLaMA2 on different tasks including MMLU, commonsense QA and etc.

**Strengths:**

- Overall, the paper is well-organized and easy to follow. The motivation is well-introduced and the method is clearly described.
- The proposed method is relatively simple, so it can be easily used for real-world application.
- The proposed method achieves competitive scores, so it makes W4A8 practically usable.

**Weaknesses:**

- The paper doesn't provide actual inference speed. It is not clear how some of expensive operations such as per-token dynamic quantization of activation affect the end-to-end performance. Even some of the layer-wise performance comparison would be useful.
- There still exist some performance gaps compared to FP16. As authors pointed, LLaMA-7B and LLaMA-13 show some degradations even compared to SmoothQuant. 2-3 MMLU score difference could be a blocker in real world applications.

**Questions:**

- It's not clear why v0 and v1 are set to 15 and 150.
- At the bottom of page 5, what are the two strategies? It's not clear in the context.
- In page 6, KV cache quantization is mentioned, but there's no more details.
- In Table 4, it's not clear what numbers are marked bold font.
- In Table 6, it seems there are variations with different calibration datasets. Is there any way to reduce the variations?

---

> ### Author Response · Authors · 2023-11-19
> **Per-token quantization adds up to 10% extra latency**
>
> $\textbf{Q1:}$ Inference speed and expensive operations.
>
> $\textbf{A1:}$  In this paper, we are primarily concerned with the theoretical upper-bound of inference time in W4A8 bit widths. The real inference speed, however, largely depends on the specific hardware support (GPU, CPU, FPGA, etc.) and the amount of engineering effort put into implementations.
>
> **Costly operations.** Below we provide an end-to-end performance comparison of per-token dynamic quantization and per-tensor quantization under the W8A8 setting. Notice that per-token dynamic quantization adds extra overhead that ranges from 5% to 10%.
>
> |Batch size |  input length | output length | FP16 |  W8A8 Per-tensor  |  W8A8 Per-token |
> | :---: | :----:| :---:| :----:| :----:| :----:|
> |1   | 1024  | 128 | 1772ms      | 1331 (1.33x)   | 1406 (1.26x) |
> |4   | 1024  | 128  |  2003ms    | 1519 (1.32x)    |  1589 (1.26x) |
> |16   |  1024  | 128  | 3163ms      | 2448 (1.29x)   | 2634 (1.20x) |
> |32   |  1024  | 128  | 4998ms      | 4013 (1.25x)   | 4492 (1.12x) |
>
> *Table i. Latency of LLaMA2-13b tested on a A100 80G GPU*
>
>
> $\textbf{Q2:}$ The degradation is shown on MMLU.
>
> $\textbf{A2:}$ It is expected that the degradation increases as we chase narrower bit widths. Noticeably, we've updated LLM-QAT W4A8 results in Table 4 while FPTQ generally prevails over it. This attests that FPTQ is a strong PTQ method that even outperforms the QAT method.  Nonetheless, the majority of our MMLU results are comparable to or better than W8A8 and W4A16, There are orthogonal ways such as weight-clipping, GPTQ, distillation, or applying better calibration data to mitigate this quantization gap, however, it is not the key focus of this paper. We maintain the current settings for a fair comparison.
>
>
> $\textbf{Q3:}$ On how v0 and v1 is selected.
>
> $\textbf{A3:}$ The values of v0 and v1 (recommended as 15 and 150) were determined empirically. We discover that per-tensor activation quantization already generates low MSE for the range (0, v0). While for (v0, v1) it calls for LAE to smooth the range first before per-tensor quantization to have an acceptable MSE. For intractable layers like FC2 that go far beyond v1, we resort to per-token quantization. According to activation distributions over a large body of LLMs in Table ii (also see Fig.5 - Fig 10 in the appendix), this selects QKV, and FC1 for LAE before per-tensor quantization, while keeping the Dense layer untouched.
>
> | Model | QKV | Dense | FC1 | FC2 |
> | :---: | :----:| :---:| :----:| :----:|
> | | min/max| min/max| min/max | min/max | min/max |
> |OPT-66B| 3.2871 / 94.87 | 0.0581/16.45 | 3.3223 / 94.81 |  0.1580 / 34.00 |
> |BLOOM-176B| 1.7451 / 97.31 | 0.0972 / 17.07 | 3.9922 / 119.81 | 0.1636 / 71.00 |
> |GLM-130B | 2.7188 / 115.12 | 0.2861 / 19.39 | 3.3125 / 124.56 | 0.5513 / 5476 |
> |LLaMA-65B | 0.4927 / 22.51 | 0.0518 / 17.40 | 0.2856 / 27.07 | 0.0485 / 887.50 |
>
> *Table ii. Activation ranges of different types of layers in various LLMs*
>
> After the application of the proposed LAE on QKV and FC1, we have a smoothed activation range that is much easier to quantize, as shown in Table iii below.
>
> | Model | QKV | Dense | FC1 | FC2 |
> | :---: | :----:| :---:| :----:| :----:|
> |OPT-66B| 0.2637 / 4.0312  | 0.0572 / 16.45 | 0.3489 / 3.2852 |  0.1570 / 34.00 |
> |BLOOM-176B| 0.2446 / 4.6680 | 0.0972 / 17.07 | 0.4021 / 6.4648 | 0.1632 / 71.06 |
> |GLM-130B | 0.4658 / 1.8955 | 0.2861 / 19.37 | 0.4006 / 2.0156 | 0.5513 / 5476 |
> |LLaMA-65B | 0.2371 / 3.1094 | 0.0518 / 17.39 | 0.1586 / 3.6973 | 0.0485 / 887.50 |
>
> *Table iii. Smoothed activation ranges (min/max) of different types of layers in various LLMs*
>
>
> $\textbf{Minor:}$
>
> - Table 4 bold numbers are mismarked and it is removed in the revision.
> - Page 5, two strategies refer to per-channel strategy and fine-grained strategy. Modern GPUs support per-block computation for tensors which can be exploited for the fine-grained strategy.
> - For the KV cache, we use per-tensor static quantization throughout all the experiments.
> - In Table 6, the variations due to different calibration datasets are actually small. Our intention was to demonstrate that our method isn't limited to specific calibration sets and that our performance remains robust even on randomly selected data.

---

> > ### Comment · Reviewer_WDjT · 2023-11-21
> >
> > Thanks for the rebuttal and revision. I would've increased the score to 7, but it's not an option. So, I will keep my score as is.

---

### Author Response · Authors · 2023-11-19
**Common Response and Revision Updates**

We deeply thank all the reviewers for your constructive and dedicated remarks. We've updated the manuscript regarding the issues mentioned in this rebuttal revision. Here is a list of updates:

- Figure 2(b) redrawn: W4A8’s time cost is made more accurate (the context decoding stage close to W8A8’s  and the self-decoding stage close to W4A16’s)
- Page 6: Added a short description for KV cache quantization. Added the reference for selecting $v_0$ and $v_1$.
- Table 3: Added SmoothQuant W4A8's result
- Table 4: Added LLM-QAT W4A8 results on MMLU and Common Sense QA. Removed previously mismarked boldface.
- Table 5: Fixed LLaMA-7B's original accuracy on HS.
- Appendix:
  - A.3 Added statistical analysis on activation ranges before and after LAE
  - A.4 Added component analysis on SmoothQuant W4A8

---

### Meta-Review · Area_Chair_745t · 2023-12-10

**Metareview:**

This paper proposes a W4A8 post-training quantization method for LLMs. While W4A8 can be practically useful for LLMs, reviewers and AC are leaning towards rejection due to limited novelty, insufficient wall-clock time justifications to W4A8, and performance degradation.

**Justification For Why Not Higher Score:**

limited novelty, insufficient wall-clock time justifications to W4A8, and performance degradation

**Justification For Why Not Lower Score:**

N/A

---

### Decision · Program_Chairs · 2024-01-16

Reject